# Optogenetic Modulation of Neural Progenitor Cells Improves Neuroregenerative Potential

**DOI:** 10.3390/ijms22010365

**Published:** 2020-12-31

**Authors:** Esther Giraldo, David Palmero-Canton, Beatriz Martinez-Rojas, Maria del Mar Sanchez-Martin, Victoria Moreno-Manzano

**Affiliations:** 1Neuronal and Tissue Regeneration Laboratory, Centro de Investigación Príncipe Felipe, 46012 Valencia, Spain; egiraldo@cipf.es (E.G.); dapalcan@etsiamn.upv.es (D.P.-C.); bmartinez@cipf.es (B.M.-R.); mmsanchez@cipf.es (M.d.M.S.-M.); 2Department of Biotechnology, Universitat Politècnica de València, 46022 Valencia, Spain

**Keywords:** spinal cord injury, neural progenitor cells, optogenetics, channelrhodopsin-2, cell therapy, neural differentiation, axon growth, astrocyte activation

## Abstract

Neural progenitor cell (NPC) transplantation possesses enormous potential for the treatment of disorders and injuries of the central nervous system, including the replacement of lost cells or the repair of host neural circuity after spinal cord injury (SCI). Importantly, cell-based therapies in this context still require improvements such as increased cell survival and host circuit integration, and we propose the implementation of optogenetics as a solution. Blue-light stimulation of NPCs engineered to ectopically express the excitatory light-sensitive protein channelrhodopsin-2 (ChR2-NPCs) prompted an influx of cations and a subsequent increase in proliferation and differentiation into oligodendrocytes and neurons and the polarization of astrocytes from a pro-inflammatory phenotype to a pro-regenerative/anti-inflammatory phenotype. Moreover, neurons derived from blue-light-stimulated ChR2-NPCs exhibited both increased branching and axon length and improved axon growth in the presence of axonal inhibitory drugs such as lysophosphatidic acid or chondroitin sulfate proteoglycan. Our results highlight the enormous potential of optogenetically stimulated NPCs as a means to increase neuroregeneration and improve cell therapy outcomes for enhancing better engraftments and cell identity upon transplantation in conditions such as SCI.

## 1. Introduction

The transplantation of neural progenitor cells (NPCs) as a treatment for central nervous system disorders such as spinal cord injury (SCI) has been widely studied [1]. Transplanted cells can promote neuroprotection [2], immunomodulation [3], and neuroregeneration [4] by secreting trophic factors, regulating inflammation, and creating new synaptic interactions that form part of host neuronal networks. Important challenges associated with NPC-based therapies remain, including poor cell survival, limited differentiation, and reduced functional engraftment, which, in turn, limits functional regeneration, offering many times discrete benefits, attributed more to early neuroprotection rather than progressive neuroregeneration due to the short transplant survival (reviewed in [2]). Many different strategies, used alone or in combination, can foster enhanced structural and functional recovery (extensively reviewed in [5]) and include growth factor cocktails [6], optimized biomaterial carriers/scaffolds [7], and small-molecule drugs that promote cell survival [8,9] or cell differentiation by inhibiting Notch signaling [10].

Optogenetic stimulation of NPCs represents a promising means of regulating neural fate and improving therapeutic outcomes [11]. Optogenetics combines optical and genetic methodologies to allow precise spatial and temporal control (at the millisecond scale) of specific cell populations [12], as opposed to classical methods such as electrical stimulation [13]. Optogenetic strategies include the ectopic expression of light-sensitive ion channels such as channelrhodopsin-2 (ChR2) which, upon stimulation by blue light, become active and trigger a passive cationic ion flow that impacts specific signaling events and influence proliferation, differentiation, survival, and/or cell death [14,15,16,17].

While optogenetic technology has become increasingly sophisticated [18], optogenetic applications in the context of cell therapy remain relatively unexplored. For example, Weick et al. employed the stable overexpression of ChR2 in embryonic stem cells to modulate neuron differentiation and synaptic plasticity after transplantation [19], showing, in addition, that the implementation of optogenetics can also help to evaluate the functional mechanisms triggered by transplanted cells at the post-synaptic inputs by in vivo loss of function [19]. Optogenetics also allows the interrogation of graft function and graft-to-host connectivity by controlling the activation of grafted cells in real-time [20,21]. Yu et al. recently established that induced pluripotent stem cell-derived NPCs possessing a novel optochemogenetics system allowed for an enhancement of neural network connections and functional outputs after transplantation in a mouse model of stroke upon daily in vivo activation [11]. Optogenetically stimulated NPCs displayed improved expression of synapsin or brain-derived neurotrophic factor and enhanced neurite outgrowth [11].

Our new study now reports that the ectopic overexpression of ChR2 in spinal cord-derived NPCs (ChR2-NPCs) allows optogenetic stimulation, which prompts an induced proliferative rate and increased maturation of functional neural cells in vitro. Optogenetic stimulation induces higher differentiation of NPCs into oligodendrocytes and neurons and the polarization of astrocytes from a pro-inflammatory phenotype to a pro-regenerative/anti-inflammatory phenotype. Additionally, neurons derived from blue-light-stimulated ChR2-NPCs, which exhibited both increased branching and axon length, improved their axon growth in the presence of axonal inhibitory drugs. We believe that our optogenetic approach will improve the regenerative potential of NPC-based therapies for various central nervous system disorders.

## 2. Results

### 2.1. Optogenetic Stimulation Increases ChR2-NPC Proliferation and Neurite Outgrowth under Proliferative Conditions

We cultured ChR2-modified NPCs (ChR2-NPCs) derived from rat embryonic spinal cords (E15) in a proliferation medium containing growth factors (including 20 ng/μL epidermal growth factor (EGF) and fibroblast growth factor (FGF)) on Matrigel coated wells. We stimulated ChR2-NPCs via exposure to 100 pulses of blue light (BL) at 470 nm (with ten seconds between each pulse) based on a previous report [21], three times a day for three consecutive days. In these conditions, we did not find any significant cytotoxic effect by analyzing the percentage of positive cells for cleaved caspase-3 (ChR2: 16 ± 5 and ChR2 + BL 14 ± 1.2). Then, we assessed proliferation in unstimulated (ChR2) and BL-stimulated ChR2-NPCs (ChR2 + BL) via immunostaining with the Ki67 proliferation marker (Figure 1A shows a representative image). The quantification of Ki67-positive cells demonstrated a significant increase in the proliferation of BL-stimulated ChR2-NPCs (*p* < 0.05, Figure 1B), thereby suggesting that the influx of cations mediated by ChR2 stimulation enhances ChR2-NPC proliferation.

We cultured ChR2-NPCs in adherent conditions in a proliferation medium before BL stimulation for three consecutive days, as previously described, to assess the effect of optogenetic stimulation on neurite outgrowth. We examined the length of extended neurites in unstimulated (ChR2) and BL-stimulated (ChR2 + BL) ChR2-NPCs via nestin staining, whose expression persists in embryonic NPCs during axonal projection (Figure 2A shows representative images) [22]. BL stimulation of ChR2-NPCs significantly increased neuritogenesis (inducing longer extended neurite-like processes) when compared to unstimulated ChR2-NPCs (*p <* 0.05, Figure 2B). Of note, unstimulated ChR2-NPCs displayed an increased number of cells lacking neurites compared to BL-stimulated ChR2-NPCs (*p* < 0.001, Figure 2C).

Taken together, these results demonstrate that in vitro optogenetic stimulation of NPC
s in proliferative conditions can significantly enhance ChR2-NPC proliferation and induce the development of neurite-like processes, with improved neuritogenesis or neurite sprouting.

### 2.2. Optogenetic Stimulation Enhances ChR2-NPC Differentiation into Neurons and Oligodendrocytes

To assess the effect of optogenetic stimulation on ChR2-NPC differentiation into neurons, oligodendrocytes, and astrocytes, we induced the spontaneous differentiation of ChR2-NPCs by growth factor withdrawal for seven days. We began BL stimulation on day three of the differentiation protocol and assessed alterations to cell differentiation through immunostaining for the neuronal marker βIII-tubulin (Figure 3A,D), the oligodendrocyte marker oligodendrocyte transcription factor 2 (Olig2) (Figure 3B,E), and the astrocyte marker glial fibrillary acidic protein (GFAP) (Figure 3C,F). We discovered that the BL stimulation of differentiating ChR2-NPCs prompted a significant increase in the percentage of βIII-tubulin-positive neurons (*p <* 0.001, Figure 3D) and oligodendrocytes with positive nuclear expression of Olig2 nuclear (*p* < 0.05, Figure 3E) but failed to influence the proportion of GFAP-positive astrocytes (Figure 3F) when compared with unstimulated differentiating ChR2-NPCs.

Overall, this suggests that the altered flux of cations into NPCs mediated by ChR2 stimulation does not influence astroglial differentiation but significantly improves neuronal and oligodendrocyte differentiation.

### 2.3. Optogenetic Stimulation of ChR2-NPCs Influences Astrocyte Maturation into Distinct Morphological and Functional Subtypes

While optogenetic stimulation did not affect the proportion of astrocytes produced during the spontaneous differentiation of ChR2-NPCs (Figure 3F), we did note significant morphological changes to astrocytes following BL stimulation. To explore this link, we quantified the number of GFAP-positive cells with polygonal or stellate morphologies (Figure 4A–D). Polygonal astrocytes possess highly-branched, densely-packed bushy processes, whereas stellate astrocytes display long, straight, and less densely-packed processes clustering two subtypes of astrocytes: polygonal, associated with a pro-inflammatory phenotype, and stellate, associated with an anti-inflammatory phenotype, as previously described [23]. We quantified the proportion of each astrocyte subtype based on their morphology and calculated the ratio over the total number of GFAP-positive cells (Figure 4A–D). Figure 4C–D demonstrate that BL stimulation of ChR2-NPCs during differentiation prompted the generation of a significantly lower proportion of polygonal astrocytes (*p* < 0.05) and a significantly higher proportion of stellate astrocytes (*p* < 0.01) as compared with unstimulated ChR2-NPCs.

To study if morphological features correspond to different functional phenotypes (pro-inflammatory or anti-inflammatory), we analyzed the expression of nuclear factor-kappa B (NF-κB) as a marker of inflammatory phenotype and Arginase-1 as a marker of anti-inflammatory phenotype [24]. Figure 4E–J demonstrate that BL stimulation of ChR2-NPCs during differentiation prompts the significant downregulation of NF-κB expression (*p* < 0.001, Figure 4E–G) and the upregulation of Arginase-1 expression (*p* < 0.001, Figure 4H–J) compared to unstimulated conditions, suggesting that cation influx during BL stimulation increases the proportion of anti-inflammatory and, hence, pro-regenerative stellate astrocytes.

### 2.4. Optogenetic Stimulation Enhances Axon Growth and Branching in ChR2-NPC-Derived Neurons

To assess whether optogenetic stimulation during the differentiation of ChR2-NPCs (Figure 3A) influences neural axon length and axon branching, we quantified both parameters in βIII-tubulin-positive neurons (Figure 5). We determined the mean number of branches per axon (Figure 5A–C) and mean axon length (Figure 5D–J) for unstimulated and BL-stimulated ChR2-NPCs after differentiation using the NeuronJ plugin. These results established that optogenetic stimulation led to longer and more abundant axonal projections, which corresponds to a more mature neuronal state.

We also evaluated the functional relevance of optogenetic stimulation during the differentiation of ChR2-NPCs on axonal growth in two inhibitory models—treatment with lysophosphatidic acid (LPA), a potent mitogen that activates the Rho/ROCK pathway and induces growth cone retraction and neurite collapse [25], or with chondroitin sulfate proteoglycans (CSPGs), which mimic one of the major inhibitory signals mediated by the glial scar [26]. Representative images of βIII-tubulin immune staining show that ChR2 activation following BL stimulation counteracted the axonal retraction induced either by LPA (Figure 5F,G) or soluble CSPGs (Figure 5H,I), showing significant axonal regrowth in both assayed models (Figure 5J).

## 3. Discussion

Stem cell therapy holds enormous potential in promoting neuroregeneration and neuroprotection in pathologies such as SCI. Grafting ChR2-NPCs provides a targeted approach to stimulate NPCs in vivo, determine their contribution to host neural circuits, and allows the activation of NPCs before and after cell transplantation to improve cell proliferation, cell differentiation, and cell plasticity. As we achieved less than 50% transduction efficiency but observed considerable effects, we hypothesize that optogenetic stimulation may enhance paracrine signaling from ChR2-NPCs into the neighboring cells, for instance, by increased secretion of neurotrophins, propagation of ion fluxes, or by increasing the release of neuro- or glio-transmitters. However, to further validate this hypothesis and the involved molecular mechanisms, additional investigation is further needed.

In the present study, we expressed a ChR2 form carrying an H134R missense mutation that induces larger photocurrents upon stimulation compared to wild-type ChR2 [27] in NPCs using the AAV9 viral vector under the control of a CMV early enhancer/chicken β actin (CAG) promoter. While we observed modest transduction efficiency, BL stimulation of ChR2-NPCs led to enhanced proliferation and neuronal and oligodendrocyte differentiation and promoted the maturation of stellate astrocytes. Previous studies have shown that optogenetic stimulation propagates astrocyte activation through a paracrine effect involving signaling mediators such as the cytokine leukemia inhibitory factor (LIF), which activates extracellular signal-related kinase (Erk) signaling and induces signal transducer and activator of transcription 3 (Stat3) activation [27]. We hypothesize that a similar paracrine effect might be also present in our BL-stimulated cultures.

BL stimulation of ChR2 gives rise to an initial cell depolarization through the influx of cations, including Na^+^, K^+^, H^+^, and Ca^2+^ [28,29]. The elevation of intracellular Ca^2+^ levels can trigger the expression of pro-survival and pro-regenerative factors by inducing several signaling pathways [11,30]. For example, the activation of calcineurin and Ca^2+^/calmodulin-dependent protein kinase II (CaMKII) by the transient elevation of intracellular Ca^2+^ modulates cell cycle progression and proliferation [31]. In our study, ChR2-NPCs displayed increased proliferation following short BL stimulation for three consecutive days compared to unstimulated ChR2-NPCs. The discrete but significant proliferative effect in stimulated ChR2-NPCs could contribute to improve cell survival rates rather than tumorigenic complications, despite the already large number of studies employing NPCs [4,7,9,11,32].

Additionally, our results regarding the effects of optogenetic stimulation on NPC differentiation and maturation revealed the potential of in vivo optogenetic stimulation after transplantation, which has been recently enabled by the development of small, wireless, and fully-internal implants for light-mediated stimulation [33,34]. Following transplantation, the lesion microenvironment induces NPCs to differentiate mostly into astrocytes, with minimal oligodendrocytic or neuronal differentiation [35,36]. Our results demonstrate that optogenetic stimulation of ChR2-NPCs during spontaneous differentiation increases the proportion of differentiated oligodendrocytes and neurons, which may improve therapeutic outcomes by promoting myelination and the formation of neuronal relay circuits, respectively. Our findings agree with previous reports that demonstrated an increase in neuronal differentiation of ChR2-expressing embryonic pluripotent stem cells following appropriate stimulation [37] and the increased neuronal differentiation and maturation of NPCs following electrical stimulation [25,38]. Ono et al. previously showed that intracellular increase in Ca^2+^ by optogenetics stimulation led to increases in the expression of GalC, an oligodendrocyte marker [39]. Thus, membrane depolarization, electrical excitability, and Ca^2+^ currents all enhance the neuronal differentiation of NPCs. ChR2-mediated depolarization of differentiating NPCs, along with the potential subsequent activation of voltage-gated calcium channels (VGCCs), could lead to an increase in intracellular Ca^2+^ concentrations that promote cyclic adenosine monophosphate (cAMP) production. The upregulation of VGCCs and the increased excitability and upregulation of pro-neuronal genes mediated by cAMP signaling-dependent pathways [40] ultimately result in increased neural differentiation of NPCs. Lepski et al. provided evidence that increased cAMP levels promote the neuronal differentiation of NPCs via the upregulation of VGCCs [41]. We discovered that optogenetic stimulation enhanced neuronal differentiation of ChR2-NPCs, with differentiated neurons displaying markedly increased axon growth and branching, suggestive of enhanced activity-dependent neuronal maturation. Moreover, neurons exhibited increased regenerative capacity when cultured in an inhibitory environment (the presence of CSPGs or following axonal retraction induced by LPA). Neuroregenerative benefits following optogenetic stimulation may result from increased intracellular Ca^2+^ concentration, whose essential role in axon growth has been experimentally demonstrated [42,43,44]. After axotomy, initial Ca^2+^ signals are crucial for growth cone formation through their participation in the cellular events that include microtubule reorganization and the reduction in membrane tension [43]. Moreover, Ca^2+^ activates cAMP/exchange protein directly activated by cAMP 2 (EPAC2) signaling, which promotes neurite outgrowth via cAMP response element-binding protein (CREB) activation [45] and phosphoinositide 3-kinase/protein kinase B (PI3K/AKT) signaling, which is essential for growth cone formation through its downstream targets mammalian target of rapamycin (mTOR), glycogen synthase kinase 3β (GSK3β), and adenomatous-polyposis-coli (APC) [46,47,48].

While we failed to observe any alteration to the proportion of astrocytes generated following the optogenetic stimulation of ChR2-NPCs during spontaneous differentiation, we did observe alterations to astrocyte morphology (polygonal to stellate) and phenotype. Early studies indicated that increases in intracellular cAMP levels could convert polygonal astrocytes into stellate astrocytes [49]. Astrocytes exhibit excitability by increasing Ca^2+^ levels, and these intracellular changes may affect astrocyte phenotype, function, and the nature and/or magnitude of gliotransmitter release. In addition to the morphological switch, astrocytes derived from BL-stimulated ChR2-NPCs expressed lower NF-ƘB levels and increased Arginase-1, suggesting that optogenetic stimulation promotes a neuroprotective astrocyte phenotype which has been shown to promote neuronal survival and tissue repair [50]. Astrocytes’ heterogeneity and their role in disease pathology/injury in the central nervous system have been extensively discussed in recent years and have revealed a role in axonal regrowth, synaptic plasticity, neuroinflammation and neuronal transmission [24,51,52].

## 4. Materials and Methods

### 4.1. NPC Isolation, Culture, ChR2 Transduction, and Photostimulation

NPCs were isolated from E-15 spinal cords from Sprague-Dawley rats dissected in ice-cold Hank’s balanced saline solution (HBSS) supplemented with penicillin-streptomycin. The dissected tissue was mechanically dissociated by repetitive pipetting, and NPCs were isolated and cultured as neurospheres in a growth medium (NeuroCult™ Proliferation Medium (Stemcell Technologies, Grenoble, France) supplemented with NeuroCult™ Proliferation Supplement (Stemcell Technologies), 100 U/mL penicillin (Sigma, St. Quentin Fallavier Cedex, France), 100 µg/mL streptomycin (Sigma), 0.7 U/mL heparin (Sigma), 20 ng/mL epidermal growth factor (EGF; Thermo Fisher, Horsham, UK), and 20 ng/mL basic fibroblast growth factor (bFGF; Invitrogen)) in ultra-low-attachment plates. For adherent growth, NPCs were cultured on Matrigel^®^-coated coverslips (diluted 1/20; Corning, Corning, NY, USA). To assess spontaneous differentiation of NPCs, neurospheres were dissociated with accutase (StemPro, Thermo Fisher) following the manufacturer’s instructions and seeded on Dulbeco’s Modified Eagle’s Medium (DMEM)/F12 supplemented with 100 U/mL penicillin, 100 μg/mL streptomycin, 2 mM L-glutamine, 5 mM HEPES buffer, 0.125% NaHCO3, 0.6% glucose, 0.025 mg/mL insulin, 80 μg/mL apotransferrin, 16 nM progesterone, 60 μM putrescine, 24 nM sodium selenite, and 2% fetal bovine serum for one week.

Ectopic expression of mCherry-tagged hChR2 (H134R) protein in NPCs was achieved using an AAV9 vector generated from the construct pAAV.CAG.hChR2(H134R)-mCherry.WPRE.SV40 (Addgene viral prep #100054-AAV9). The DNA construct delivered to transduced cells carries a codon-optimized gene encoding the BL-activated unspecific cation channel ChR2 with the missense mutation H134R, enabling larger photocurrents upon BL stimulation compared with wild-type ChR2 [53]. For transduction of NPCs, neurospheres were dissociated with accutase, suspended in 200 µl of culture medium containing concentrated virus (10^5^ transducing units/cell), and then incubated at 37° C for 1 h. NPCs were then plated onto ultra-low-attachment plates and allowed to grow for seven days for neurosphere formation. At seven days after transduction, transduction efficiency was evaluated by flow cytometry, showing successful ChR2-mCherry expression in 42% ± 9.6 of live NPCs (data not shown). Photostimulation was applied to the BL-stimulated group using a PerkinElmer Ensight multimode plate reader (PerkinElmer SL, Madrid, Spain). The stimulation pattern consisted of three BL exposures (100 pulses at 470 nm) separated by 10 s, which efficiently induced Ca^2+^ influx assayed using a Fluor4 probe (Thermo Fisher; data not shown), for three consecutive days (study of optogenetic stimulation on proliferation) or five days (study of differentiation). All assayed parameters included a control of non-transduced NPCs subjected to BL stimulation, and no differences were observed between non-stimulated ChR2-NPCs and non-stimulated non-transduced NPCs.

### 4.2. Drug Administrations

In order to induce in vitro neurite retraction, cultures were treated with LPA (L726, Sigma). LPA solution was reconstituted in sterile Milli-Q Ultrapure water at a stock concentration of 2 mM. Neurite retraction was induced at day 3 of the differentiation protocol by adding LPA at a final concentration of 10 µM.

In order to mimic inhibitory signals of the glial scar, 2 µg/mL of a mixture of extracellular CSPGs purified from embryonic chicken brains (Millipore, Burlington, MA, USA; provided as a liquid in PBS) was added to the cultures at day 3 of the differentiation assay.

### 4.3. Immunocytochemical Staining, Image Acquisition, and Quantification

Cultured NPCs were fixed with 4% paraformaldehyde in PBS for 10 min, then permeabilized and blocked with 5% normal goat serum (NGS; Thermo Fisher) and 0.2% Triton X-100 (Sigma). Cells were incubated overnight at 4 °C with primary antibodies against cleaved caspase-3 (1:400 9664, Cell Signaling Technology, CA, USA), Ki67 (1:400 ab15580, Abcam, Cambridge, UK), nestin (1:400 ab6142, Abcam), β-III-tubulin (1:500 MO15013, Neuromics, Edina, MN, USA), Olig2 (1:500 ab33427, Millipore), GFAP (1:1000 PA1-10004, Thermo Fisher), NF-ƘB (1:400 SC-8008, Santa Cruz, CA, USA), and Arginase-1 (1:400 SC-271430; Santa Cruz). For secondary antibodies, either AlexaFluor-488, -555, or -647 (1:400, Invitrogen) conjugated antibody against the respective IgG was used. DAPI was used to stain cell nuclei.

Fluorescent images were acquired using a fluorescent microscope (Apotome; Zeiss, Oberkochen, Germany) or a confocal microscope (Leica, Wetzlar, Germany). Consistent exposures were applied for all images, and ImageJ software was used for image analysis. The NeuronJ plugin was used to quantify neurite and axon lengths.

### 4.4. Statistical Analyses

Data are presented as mean ± standard error of the mean (SEM) and were analyzed using GraphPad Prism Software. The Shapiro–Wilk normality test was performed to ensure normal data distribution. Comparisons between two groups and among multiple groups used the unpaired Student’s *t*-test and one-way ANOVA, respectively, using Bonferroni’s multiple comparison test for post-hoc analysis. If normality was not met, non-parametric tests were used (the Mann–Whitney rank-sum test). *p* values of ≤ 0.05 were considered statistically significant.

## 5. Conclusions

Our findings provide evidence that the in vitro optogenetic stimulation of ChR2-NPCs increased NPC proliferation and oligodendrocyte and neuronal differentiation and enhanced neurite growth in resultant neurons as well as the polarization of astrocytes into a neuroprotective phenotype. However, in order to evaluate whether the outcomes from this study can improve cell therapy and promote neuro-regeneration/-protection post-transplantation, further studies evaluating the effects of transplanting optogenetic-stimulated NPCs in an in vivo model of neurodegeneration are needed.

## Figures and Tables

**Figure 1 ijms-22-00365-f001:**
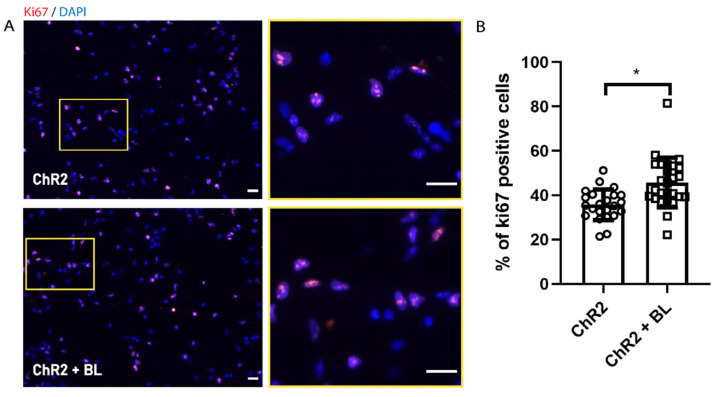
Optogenetic stimulation increases channelrhodopsin-2 (ChR2) in spinal cord-derived NPC (ChR2-NPC) proliferation. (**A**) Representative immunofluorescence images for the proliferation marker Ki67 (red) with nuclear staining by 4′,6-diamidino-2-phenylindole (DAPI) (blue) in unstimulated (upper panels) and blue light (BL)-stimulated ChR2-NPCs (lower panels). Yellow square: Inset showing higher magnification. (**B**) Quantification of Ki67-positive ChR2-NPCs expressed as the percentage of the total number of cells in unstimulated NPCs (ChR2) or BL-stimulated NPCs (ChR2 + BL) for three consecutive days. Data expressed as mean ± SEM from three independent experiments (ChR2: 34.8 ± 1.2; ChR2 + BL: 43.8 ± 2.3). * *p* < 0.05 vs. ChR2 determined by unpaired *t*-test. Scale bars, 20 µM.

**Figure 2 ijms-22-00365-f002:**
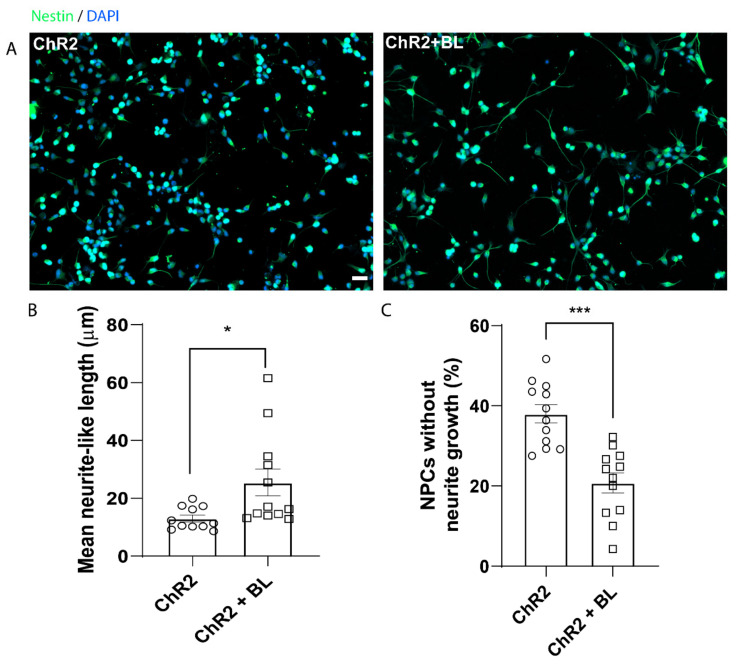
Optogenetic stimulation prompts the appearance of neurite-like processes in ChR2-NPCs. (**A**). Representative images of immunofluorescence staining for Nestin (green) and nuclear detection by DAPI (blue) in unstimulated (ChR2) and BL-stimulated ChR2-NPCs (ChR2 + BL). (**B**). Quantification of mean length of neurite-like processes. The most extended process from each NPC was measured using Neuron-J. Data expressed as mean ± SEM from three independent experiments (ChR2: 13 ± 1.2; ChR2 + BL: 25.4 ± 4.6). * *p* < 0.05 vs. ChR2 determined by unpaired *t*-test. (**C**) Quantification of the percentage of ChR2-NPCs that failed to develop neurite-like processes. Data expressed as mean ± SEM from three independent experiments (ChR2: 38 ± 2.3; ChR2 + BL: 20.8 ± 2.5). *** *p* < 0.001 vs. ChR2 determined by unpaired *t*-test. Scale bar, 20 µM.

**Figure 3 ijms-22-00365-f003:**
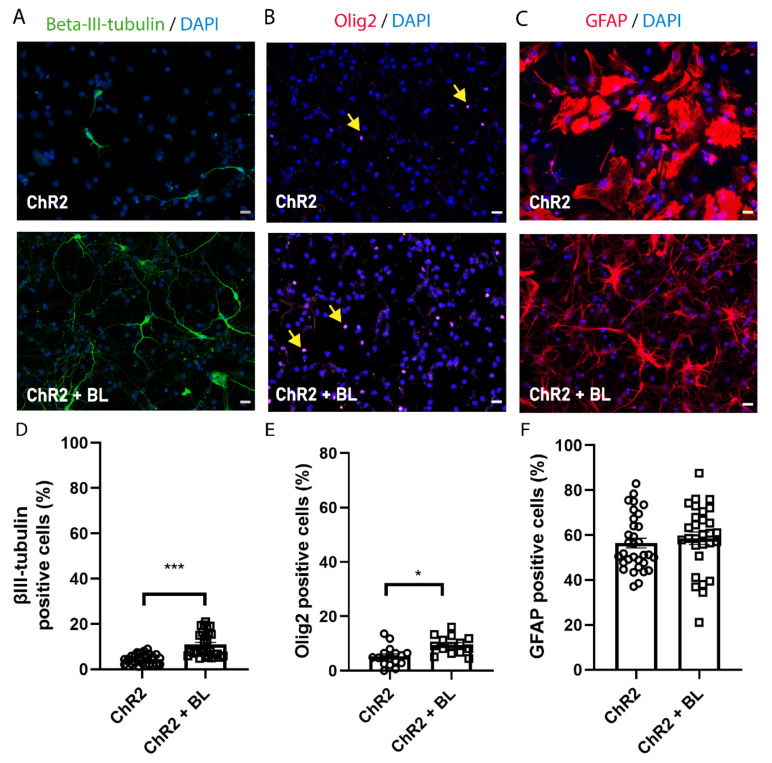
Optogenetic stimulation enhances neuronal and oligodendrocytic but not astrocytic differentiation of NPCs. ChR2-NPCs were differentiated for seven days. (**A**–**C**) Representative images of immunofluorescence staining for beta-III-tubulin (green; A), oligodendrocyte transcription factor 2 (Olig2) (red; nuclear signal indicated with yellow arrows; B), and glial fibrillary acidic protein (GFAP) (red; C) for unstimulated (ChR2) and BL-stimulated ChR2-NPCs (ChR2 + BL). (**D**) Quantification of the percentage of neurons (beta-III-tubulin-positive cells). Data are expressed as mean ± SEM from three independent experiments (ChR2: 4.5 ± 0.4; ChR2 + BL: 10.9 ± 1) *** *p* < 0.001 vs. ChR2 determined by Mann–Whitney test. (**E**) Quantification of the percentage of oligodendrocytes (Olig2-positive cells). Data are expressed as mean ± SEM from three independent experiments (ChR2: 5.3 ± 0.9; ChR2 + BL: 9.5 ± 0.9) * *p* < 0.05, unpaired *t*-test. (**F**) Quantification of the percentage of astrocytes (GFAP-positive cells). Data are expressed as mean ± SEM from three independent experiments (ChR2: 56.47 ± 2.2; ChR2 + BL: 58.65 ± 2.7). Scale bars, 20 µM.

**Figure 4 ijms-22-00365-f004:**
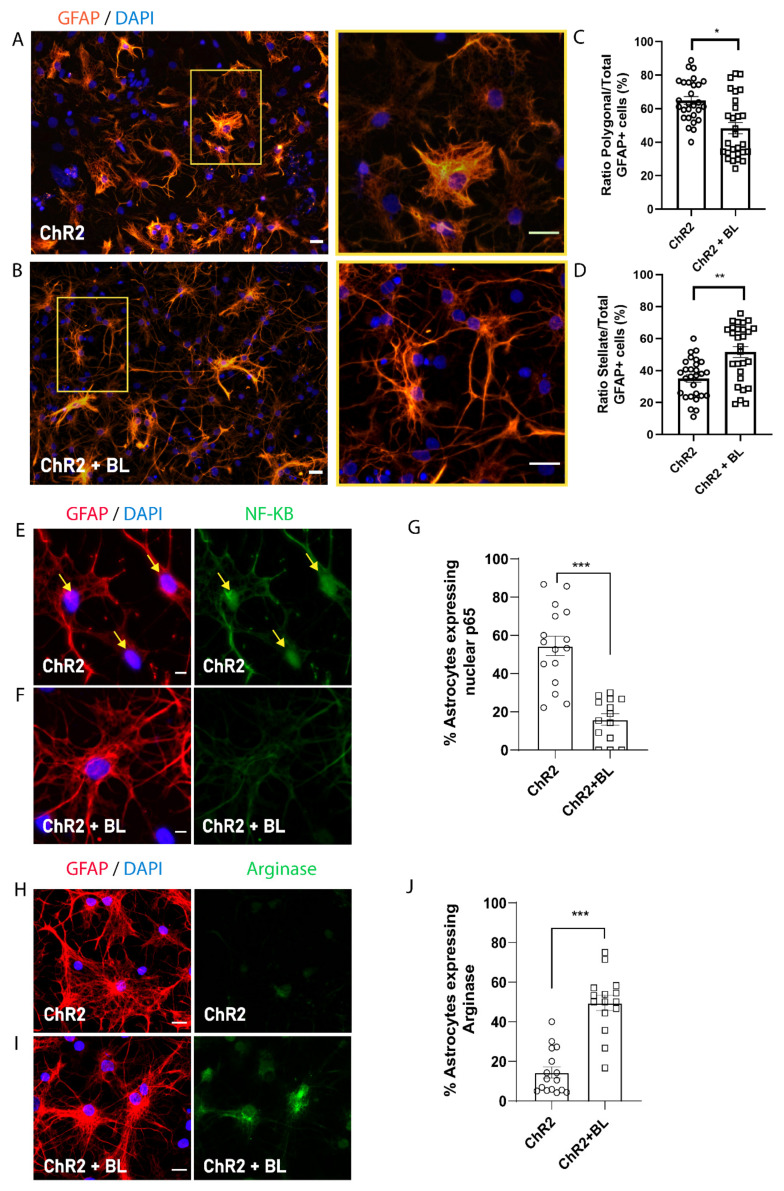
Optogenetic stimulation triggers morphological and functional shift in ChR2-NPC-derived astrocytes. Astrocyte morphology was determined after GFAP staining. (**A**) Representative images of immunofluorescence staining for GFAP (orange) and nuclear detection by 4′,6-diamidino-2-phenylindole (DAPI) (blue) in unstimulated ChR2-NPCs (ChR2). Yellow square shows higher magnification for polygonal astrocytes. Scale bars, 20 µM. (**B**) Representative images of immunofluorescence staining for GFAP (orange) and nuclear detection by DAPI (blue) in BL-stimulated ChR2-NPCs (ChR2 + BL). Yellow square shows higher magnification for stellate astrocyte. Scale bars, 20 µM. (**C**) Quantification of the ratio of polygonal astrocytes in unstimulated (ChR2) and BL-stimulated ChR2-NPCs (ChR2 + BL) cultures. Data are expressed as mean ± SEM from three independent experiments (ChR2: 64.9 ± 2.3; ChR2 + BL: 48.4 ± 3.4). * *p* < 0.05 vs. ChR2 determined by unpaired *t*-test. (**D**) Quantification of the ratio of stellate astrocytes in unstimulated (ChR2) and BL-stimulated ChR2-NPCs (ChR2 + BL) cultures. Data are expressed as mean ± SEM from three independent experiments (ChR2: 35.1 ± 2.3; ChR2 + BL: 51.6 ± 3.4). ** *p* < 0.01 vs. ChR2 determined by unpaired *t*-test. (**E**) Representative images of immunofluorescence staining for GFAP (red), nuclear factor-kappa B (NF-kB) (green), and nuclear detection by DAPI (blue) in unstimulated ChR2-NPCs (ChR2). Yellow arrows show nuclear staining. Scale bars, 10 µM. (**F**) Representative images of immunofluorescence staining for GFAP (red), NF-kB (green), and nuclear detection by DAPI (blue) in BL-stimulated ChR2-NPCs (ChR2 + BL). Scale bars, 10 µM. (**G**) Quantification of the percentage of astrocytes expressing nuclear NF-kB among unstimulated and BL-stimulated groups. Data are expressed as mean ± SEM from three independent experiments (ChR2: 54.5 ± 5.1; ChR2 + BL: 15.9 ± 3). *** *p*< 0.001 vs. ChR2 determined by unpaired *t*-test. (**H**) Representative images of immunofluorescence staining for GFAP (red), Arginase-1 (green), and nuclear detection by DAPI (blue) in unstimulated ChR2-NPCs (ChR2). Scale bars, 10 µM. (**I**) Representative images of immunofluorescence staining for GFAP (red), Arginase-1 (green), and nuclear detection by DAPI (blue) in BL-stimulated ChR2-NPCs (ChR2 + BL). Scale bars, 10 µM. (**J**) Quantification of the percentage of astrocytes expressing Arginase-1 among unstimulated and BL-stimulated groups. Data are expressed as mean ± SEM from three independent experiments (ChR2: 14.5 ± 2.8; ChR2 + BL: 49.6 ± 3.9). *** *p* < 0.001 vs. ChR2 determined by unpaired *t*-test.

**Figure 5 ijms-22-00365-f005:**
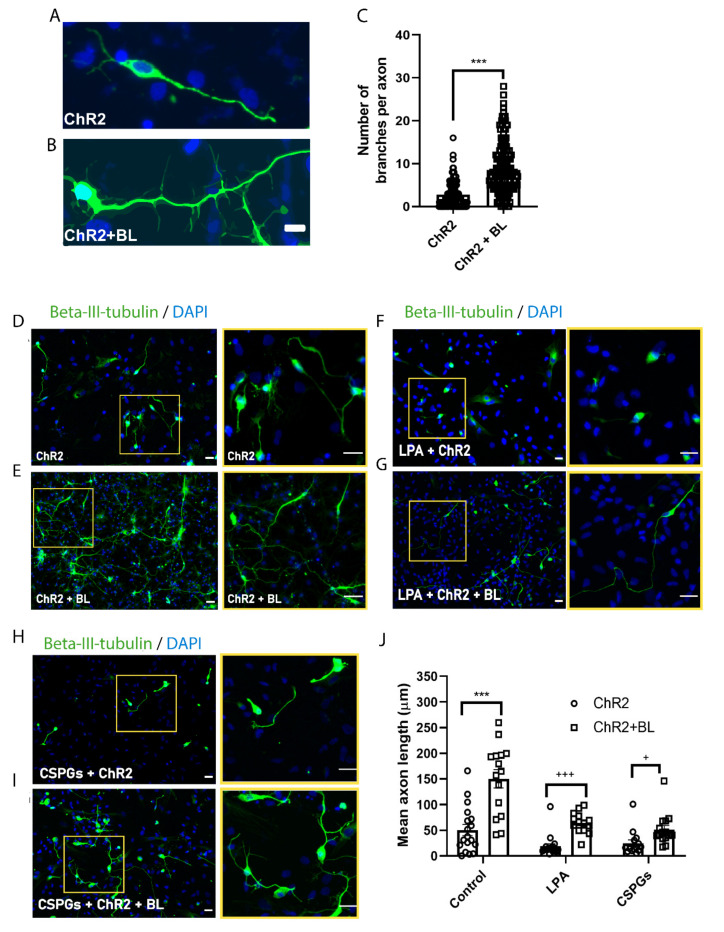
ChR2 activation promotes axonal branching and growth in ChR2-NPC-derived neurons and counteracts pharmacologically-induced axonal retraction. (**A**,**B**). Representative images of immunofluorescence staining for beta-III-tubulin (green) and nuclear detection by DAPI (blue) in unstimulated ChR2-NPCs (ChR2) (A) and BL-stimulated ChR2-NPCs (ChR2 + BL) (B). Scale bars, 10 µM. (**C**) Quantification of branches per axon in unstimulated (ChR2) and BL-stimulated ChR2-NPCs (ChR2 + BL). Data are expressed as mean ± SEM (ChR2: 2.8 ± 0.2; ChR2 + BL: 8.5 ± 0.4). *** *p* < 0.001 vs. ChR2 determined by unpaired *t*-test. (**D**,**E**) Representative images of immunofluorescence staining for beta-III-tubulin (green) and nuclear detection by DAPI (blue) in unstimulated ChR2-NPCs (ChR2) (D) and in BL-stimulated ChR2-NPCs (ChR2 + BL). (E). Yellow squares show higher magnification of selected neurons. Scale bars, 20 µM. (**F**,**G**) Representative images of immunofluorescence staining for beta-III-tubulin (green) and nuclear detection by DAPI (blue) in unstimulated (ChR2) (D) and BL-stimulated ChR2-NPCs (ChR2 + BL) (E) in the presence of LPA (10 μM). Yellow squares show higher magnification of selected neurons. Scale bars, 20 µM. (**H**,**I**). Representative images of immunofluorescence staining for beta-III-tubulin (green) and nuclear detection by DAPI (blue) in unstimulated (ChR2) (H) and BL-stimulated ChR2-NPCs (ChR2 + BL) (I) in the presence of chondroitin sulfate proteoglycans (CSPGs) (2 µg/mL) at day three of differentiation. Yellow squares show higher magnification of selected neurons. Scale bars, 20 µM. (**J**) Quantification of mean axonal length for untreated (ChR2: 50.6 ± 10.7; ChR2 + BL: 150.4 ± 17.8), LPA-treated (ChR2: 18.9 ± 4.8; ChR2 + BL: 69.7 ± 6.6), and CSPG-treated (ChR2: 24.9 ± 6.6; ChR2 + BL: 51.3 ± 8) unstimulated (ChR2) and BL-stimulated ChR2-NPCs (ChR2 + BL). Data are expressed as mean ± SEM from three independent experiments. *** *p* < 0.001 vs. ChR2 determined by unpaired *t*-test; ^+++^
*p* < 0.001; ^+^
*p* < 0.05 vs. ChR2 determined by Mann–Whitney test. Scale bars, 20 µM.

## Data Availability

Data is contained within the article.

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
