# Peer review of "Optogenetic Modulation of Neural Progenitor Cells Improves Neuroregenerative Potential"

_ijms, 2020, doi:10.3390/ijms22010365_

Round 1

Reviewer 1 Report

In this work, the effect of the in vitro optogenetic stimulation of ChR2-NPCs is studied. The results presented here shows that optogenetic stimulus of ChR2-NPCs promotes the proliferation of progenitors and differentiation into neurons, oligodendrocytes, as well as a more “beneficial” conformation of astrocytes. An important challenge associated with NPCs-based therapies is the limited differentiation of this cells. This work could be the basis for studies on the beneficial effect of optogenetic stimulation in conjunction with NPCs transplantation to promote neuroregeneration.

Overall, the work is well presented and the results are well discussed. I have some questions/suggestions for the authors that could improve it

  • Why did the authors apply “three BL exposures (100 pulses at 470 nm) separated by 10 s for three consecutive days (study of optogenetic stimulation on proliferation) or five days (study of differentiation)”. On what were they based to select these stimuli?
  • Regarding the graphs presented in figures1B; 2B, C; 3D, E, F, 4C, D, G, J, 5C, H: It would be better to present data in a manner that more transparently shows the sample size and distribution. Swarm plots, for example, are greatly preferable to bar graphs. (See PLoS Biol. 2015; 13:e1002128. doi: 10.1371/journal.pbio.1002128). Moreover, data expressed as the Mean ± SEM but also, the Mean ± SEM of each group should be indicated in the figure footnotes because it is not easy to extract the value from the graph.
  • Related to proliferation data, did the authors see that there was the same difference in proliferation on all the days that they made the counts, or did the effect of the stimulation on proliferation increase as it is repeated? (% of proliferation first day vs % proliferation third day). This could give an idea if a single dose BL exposures is sufficient or repeated stimulation increases the effect.
  • In figure 2A, the scale value is not indicated and one of the images has no scale.
  • In figure 3B, in addition to the characteristic Olig2 pattern, in the image of the ChR2 + BL group, there is at least 1 cell that seems to express it also in the cytoplasm and neurites (located between the two yellow arrows). How do the authors explain this?
  • Materials and Methods could be more orderly and clear if the drug administration method is written in a different section. Moreover, the authors should indicate the solvent in which the drugs were dissolved.
  • In the conclusions, the authors discuss the possibility that the efficiency of optogenetic stimulation could be increased in a paracrine way. This is a very interesting hypothesis, and future studies should check it, but it is not a conclusion from this work.
  • The results shown in this work are very interesting and promising. It is demonstrated that in vitro optogenetic stimulation of ChR2-NPCs improved NPCs proliferation, increased oligodendrocyte and neuronal differentiation, enhanced neurite growth in resultant neurons, and the polarization of astrocytes into a neuroprotective phenotype, however, this does not make evident that optogenetic stimulation of ChR2-NPCs promotes neuro-regeneration/-protection post-transplantation. Therefore, the authors should be more cautious in their conclusions.

I hope my suggestions help the authors to improve the manuscript.

Reviewer 2 Report

The manuscript by Giraldo et al examines the positive effect of optogenetic stimulation of spinal cord-derived neural progenitor cells on proliferation and differentiation in vitro.  The authors also demonstrate that the NPCs have improved axon growth when challenged with drugs that typically inhibit neurite outgrowth. These findings are very interesting and may have implications for cell therapy.  There are a few issues that should be addressed in order to strengthen the study.

  1. Is there a control group of mock transfection or transfection with a non-functional ChR2 followed by BL exposure to ensure that it is not the BL that is causing the observed effects?

  1. Is it known if the proliferating and differentiating cells are the ones that were transduced? In other words, can the mCherry marker expression be correlated with Ki67 or neurites?  Given that the transduction rate of less than 50% of the cells, this is something that should be considered and would support the argument of autocrine vs paracrine effects. 

  1. Have the authors counted the total number of cells before and after BL exposure to know whether there is an effect on total cell counts?

  1. Similarly, has a marker for apoptosis such as activated Caspase-3 been examined to know if there is an effect on cell death and whether the proliferating cells are surviving?

  1. Is the low proportion of BetaIII tubulin+ cells and high proportion of GFAP+ cells following differentiation protocol (ie removal of growth factors) expected? What do the authors think would happen if a protocol that favored neuronal differentiation over astrocytic differentiation were applied? 

  1. What do the authors mean by independent experiments. Were they started from new NPC cultures from separate rats or derived from a new neurosphere culture of the same rat?

  1. How many dishes were the 5 fields obtained from for data analysis?

  1. Is there any evidence of an actual increase in activity as a result of the Blue Light? Is there a way to demonstrate that the observed effects on proliferation and differentiation are a direct result of the increased cation flow?

  1. Is there a concern for in vivo application of over proliferation or improper integration into circuitry that could cause either tumor formation or seizures, respectively? This should be addressed in the Discussion?

  1. Figure 5B should include a better example of increased axonal branching and growth. It is not obvious that there is a change from -BL at high magnification.

  1. The language could be slightly improved (e.g. typo on line 48 should be unexplored)

Round 2

Reviewer 2 Report

The authors have addressed all of the concerns of this reviewer.